# Metabolically healthy obesity: Inflammatory biomarkers and adipokines in elderly population

**Lidia Cobos-Palacios[1], María Isabel Ruiz-Moreno[1], Alberto Vilches-Perez[2], Antonio Vargas-Candela[1], Mónica Muñoz-Úbeda[1], Javier Benítez Porres[3], Ana Navarro-Sanz[4], María Dolores Lopez-Carmona[1], Jaime Sanz-Canovas[1], Luis M. Perez-Belmonte[1], Juan José Mancebo-Sevilla[1], Ricardo Gomez-Huelgas[1,5]\*, María Rosa Bernal-Lopez[1,5]\***

1 Department of Internal Medicine, Instituto de Investigacion Biomedica de Malaga (IBIMA), Regional University Hospital of Malaga, University of Malaga, Malaga, Spain, 2 Department of Endocrinology and Nutrition, Instituto de Investigacion Biomedica de Malaga (IBIMA), University Hospital Virgen de la Victoria, Malaga, Spain, 3 Physical Education and Sports Area, Faculty of Medicine, University of Malaga, Malaga, Spain, 4 Sports Area, Sport Medicine, Malaga City Hall, Malaga, Spain, 5 CIBER Fisiopatología de la Obesidad y la Nutrición, Instituto de Salud Carlos III, Madrid, Spain

\* rosa.bernal@ibima.eu (MRBL); ricardogomezhuelgas@hotmail.com (RGH)

**Data Availability Statement:** Data cannot be shared publicly because of confidential data. Data are available from the IBIMA Institutional Data

## Abstract

### Background and aims

Obesity is linked to elevated levels of inflammatory serum markers such as C-reactive protein (CRP), interleukin-6 (IL-6), and tumor necrosis factor alpha (TNFa). Adiponectin and resistin are adipokines related to obesity. It has been described that adipose tissue presents a high production and secretion of these diverse pro-inflammatory molecules, which may have local effects on the physiology of fat cells as well as systemic effects on other organs. Our aim was to evaluate the impact that lifestyle modifications, by following a Mediterranean Diet (MedDiet) program and physical activity (PA) training, would have on inflammatory biomarkers and adipokine profile in a Metabolically Healthy Obese (MHO) elderly population from Malaga (Andalusia, Spain).

### Subjetcs and methods

Subjects aged ≥65 years (65 to 87 years old) with obesity (BMI ≥30 kg/m$^2$) were included in this study if they met ≤1 of the following criteria: systolic blood pressure ≥130 mmHg and/or diastolic blood pressure ≥ 85 mmHg; triglycerides ≥150 mg/dL; HDL-C <40mg/dL in men and <50mg/dL women; and fasting blood glucose ≥100mg/dL. Selected subjects underwent a personalized intensive lifestyle modification. Anthropometric measurements, PA, MedDiet adherence, analytical parameters, and inflammatory biomarkers were analyzed after 12 months of intervention.

### Results

166 MHO elderly subjects, 40 (24.1%) male and 126 (75.9%) female (p < 0.0001), aged 71.7±5.2 years old (65 to 87 years old) were included in the study. After 12 months of

Access / Ethics Committee (contact via D. Andres Gonzalez Jimenez, bioinformatica@ibima.eu) for researchers who meet the criteria for access to confidential data.

**Funding:** This study was funded by grants from the Instituto de Salud Carlos III, cofunded by the Fondo Europeo de Desarrollo Regional-FEDER, PI18/00766 and "Centros de Investigación En Red" (CIBER, CB06/03/0018). Authors were not specific recipients of funding. Instituto de Salud Carlos III, cofunded by the Fondo Europeo de Desarrollo Regional-FEDER Funding was also provided to María Rosa Bernal-López, who was supported by "Miguel Servet Type II" program (CPII/00014) and "Nicolás Monardes" program (C1-0005-2020) supported by Consejería de Salud, Junta de Andalucía. Lidia Cobos-Palacios and Jaime Sanz-Cánovas were supported by "Rio Hortega" program (CM20/00125 and CM20/00212, respectively), from the ISCIII-Madrid (Spain), cofunded by the Fondo Europeo de Desarrollo Regional-FEDER. And Mónica Muñoz-Úbeda was supported by Consejeria de Salud, Junta de Andalucía (RH-0100-2020).

**Competing interests:** The authors have declared that no competing interests exist.

intervention, only the waist circumference was significantly reduced in all the population (-2.5 cm, p<0.0001), although weight and BMI were maintained. MedDiet adherence increased significantly (p<0.001), but all intensity levels of PA decreased significantly (p<0.001). Concerning inflammatory biomarkers, only TNFa serum increased their levels after the intervention (p<0.001). Regarding the adipokine profile, adiponectin concentrations experienced a significant increment (p<0.001); besides, resistin concentrations decreased significantly (p<0.001). In this sense, only TNFa, adiponectin, and resistin correlated with PA. Adiponectin also correlates with insulin, triglycerides and HDL-c in baseline conditions and after 12 months of intervention; CRP, IL-6, TNFa, adiponectin, and resistin concentrations correlated with anthropometric parameters and some intensities of PA. In addition, adiponectin levels correlates with insulin, triglycerides and HDL-c. In baseline conditions, resistin levels correlated positively with TNFa (p = 0.01) and CRP (p<0.0001) levels. TNFa and IL-6 correlated positively with CRP (p = 0.03 and p<0.0001, respectively). After 12 months of intervention, only IL-6 correlated positively with CRP (p = 0.006). In addition, adipokines levels correlated positively during the process of lifestyle modification. However, during this process, only IL-6 correlated positively with itself (p<0.0001) and with CRP (p = 0.03).

## Conclusion

Healthy aging is a multifactorial biological process in which lifestyle is essential. The presence of obesity in elderly metabolically healthy population is not a problem necessarily. Elderly MHO population who eat a MedDiet and practice regularly PA are capable to modulate their production of inflammatory cytokines (CRP, IL-6, TNFa) and adipokines profile (adiponectin, resistin), preventing other metabolic disorders.

## Introduction

Obesity is a complex heterogeneous syndrome that can be defined as a multifactorial chronic disease characterized by an increase in adipose tissue. It is influenced by metabolic, physiological, genetic, cellular, molecular, cultural, and social factors [1]. Obesity is considered as an important public health problem in both developed and underdeveloped countries. The World Health Organization (WHO) has estimated that, by the end of 2030, there will be approximately 370 million people suffering from obesity and its associated comorbidities worldwide [2]. However, some individuals with obesity do not suffer from cardiometabolic complications such as dyslipidemia, diabetes, hyperuricemia, or hypertension. This phenotype is called metabolically healthy obese (MHO). The MHO phenotype shows a favorable metabolic profile characterized by high insulin sensitivity, low hepatic steatosis, normal blood pressure, low visceral adiposity, less infiltration of macrophages into adipose tissue, smaller adipocyte cell size, and favorable lipid, inflammatory and hormonal profiles. However, some studies have observed similar risk of cardiovascular events and all-cause mortality among MHO individuals compared to "metabolically unhealthy obese" (MUHO) and "metabolically healthy normal-weight" phenotypes (MHNW).

Among all metabolic disorders caused by obesity, insulin resistance (IR) is one of the most important as it is associated with a chronic state of subclinical inflammation, characterized by increased serum concentrations of C-reactive protein (CRP) [3], interleukin (IL)-6, IL-8,

monocyte chemotactic protein (MCP)-1 [4], tumor necrosis factor (TNFa) [5], amyloid A, resistin, leptin, and adiponectin [6, 7]. Changes in inflammatory signaling by adipocytes and infiltration of adipose tissue by immune cells are key features of obesity-induced IR and several associated metabolic complications. Moreover, the levels of adipokines are usually increased in people with obesity, and their circulating concentrations increase with the degree of obesity [8]. Tipping the inflammation balance in adipose tissue might be particularly important for metabolic health.

It is well known that lifestyle factors such as diet and physical activity (PA) affect inflammation profile and contribute to the functionality of adipose tissue [9]. Furthermore, food diet modifications along with a regular PA can lead to weight loss. The Mediterranean diet (Med-Diet) is a dietary pattern characterized by high consumption of fruits, vegetables, legumes and olive oil as the main source of fat intake, moderate-to-high fish intake, and moderate-to-low consumption of meat and poultry [10]. MedDiet adherence is associated with lower risk of developing type-2 diabetes and cardiovascular diseases [11]. Regular PA in the elderly improves health in many ways and is essential for healthy aging. PA helps to preserve physical function and mobility, which directly translates into a longer independence maintenance.

The mechanisms that could explain this benign profile of MHO subjects are still unclear. Obesity and its associated diseases have been deeply studied. However, data from MHO population are limited, particularly from MHO elderly population [12]. Some evidence suggests that factors such as visceral fat deposition pattern, birth weight, adipocyte size, or the expression of certain genes that regulate the differentiation and expansion of adipose tissue could be involved [13]. Currently, research on the MHO phenotype is of great interest as it can provide key data for a better understanding of the pathophysiological mechanisms of obesity. The purpose of the current study was to analyze the impact that 12 months of lifestyle modifications, by following a MedDiet program and daily PA training, would have on inflammatory biomarkers (CRP, IL-6, TNF-α) and adipokine profile (adiponectin and resistin) in a MHO elderly Spanish population from Malaga (Andalusia).

## Subjects and methods

### Study design and inclusion criteria

This is a cross-sectional study of a MHO elderly population that includes participants of both sexes to promote a lifestyle modification (with MedDiet and PA) during a total period of 12 months. Inclusion criteria were men and women aged ≥65 years with obesity (BMI ≥30-<40 kg/m$^2$) and 1 or none of the following 4 cardio-metabolic disorders: systolic blood pressure ≥140 mmHg and/or diastolic blood pressure ≥ 90 mmHg; triglycerides ≥150 mg/dL; HDL-C <40mg/dL in men and <50mg/dL women; and fasting blood glucose ≥100mg/dL, following the WHO criterion of MHO. Patients were excluded from the study if they had more than one criteria of metabolic disorders or one of the following diseases: diabetes, hypertension, previous cardiovascular disease (coronary, cerebrovascular or peripheral; aortic aneurysm, heart failure); severe associated disease (advanced organ failure, dementia, cancer); immobilized or terminally ill individuals; alcoholism or drug addiction; severe psychiatric illness; weight loss ≥5 kg of unknown cause in the last 6 months.

### Procedure

The recruitment period was from November 2018 to March 2020. It was carried out via visits performed to centers for the healthy elderly belonging to the Sports Area (Sport Medicine) of Malaga City Hall (Andalusia, Spain). Once the possible participants were selected, they were contacted and informed about the study design and objectives and summoned to the

Department of Internal Medicine in the Regional University Hospital of Malaga. All participants in the study gave their written informed consent and protocols were approved by the institutional ethics committee (Comité de Ética de la Investigación Provincial de Malaga (ref: PI18/00766-260718), belonging to the Andalusian Health Service). This study is registered with the International Standard Randomised Controlled Trial Number Register (ISRCTN) with ID ISRCTN11769612. https://www.isrctn.com/ ISRCTN11769612. Registration date: 29 November 2021.

## Intervention

After having evaluated eating behaviors, participants were encouraged to carry out lifestyle modifications to promote a healthy diet, as well as daily aged-adapted PA, which was recommended following the internationally-accepted PA guidelines [14].

The MedDiet recommended to the participants included extra virgin olive oil and nut consumption. The recommended caloric intake was 1500–1750 kcal/day, distributed as follows: 30% of fats (5–8% of saturated fatty acids, 15–18% of monounsaturated fatty acids, 5–8% of polyunsaturated fatty acids and <300 mg of cholesterol/day), 55% of carbohydrates (<10% of simple sugars, 40% complex sugars and low glycemic index), and 15% of proteins [15].

## Visits

The subjects who participated in the study had a visit at the beginning of the study and after 12 months of intervention. Anthropometric measurement (weight, height, BMI, waist circumference, and blood pressure), lifestyle questionnaires and blood samples extractions were obtained by trained sanitary personnel in both visits. Food intake questionnaires and PA measured by acelerometry were also included in order to analyze impact of the the lifestyle intervention.

## Assays

Anthropometric parameters were measured by trained nurses according to the protocol. Weight was measured using an electronic scale (TANITA Body Composition Analyzer. Type TBF-300 MA. TANITA Corporation; 1-14-2 Maeno-cho, Itabashi-ku. Tokio, Japan). Height was measured with no shoes using a wall stadiometer (Stadiometer Barys Electra Model. 511-300-A0A. ASIMED). BMI ($kg/m^2$) was calculated as weight (kg) divided by height (m) squared. The waist/hip index (WHI) was calculated as the ratio of abdominal circumference (at the level of the mid-point between the anterosuperior iliac crest and the last costal arch, parallel to the ground and upon exhalation) and hip, both in cm. Systolic blood pressure (SBP) and diastolic blood pressure (DBP) were calculated as the mean of three measurements after a 5-minute rest and measured using an automated electronic sphygmomanometer (OMRON M7; HEM-780-E).

Food intake was analyzed using questionnaires. A non-consecutive, 3-days dietary record (two workdays and one weekend day), containing detailed information about food composition and cooking recipes over 72 hours [16] and a food frequency questionnaire (number of times/day, number of days/week, number of days/14 days, number of days/month, rarely, or never) were completed in every visit [17]. Adherence to MedDiet was assessed by a validated 14-item food consumption frequency questionnaire, according to Trichopoulou A et al. [10]. 12–14 points was considered high MedDiet adherence, 8–11 points was considered moderate adherence, 5–7 points was considered low adherence, and <5 points was considered very low adherence.

PA was also monitored. Subjects were given GENEActiv Actigraph GT3X+ accelerometers to collect several data. The accelerometer should be worn under the chest with a tight elastic belt to ensure close contact with the body. Recordings were made every day for at least 7 days (weekdays and the weekend) to assess their hours of PA and sleep, except during water activities.

Blood samples were taken after an overnight fast. Besides, blood samples were processed to obtain blood serum, which was aliquoted and stored at -80˚C until its use. Biochemical levels (inflammatory biomarkers and adipokines profile) were measured in the Biomedical Research Laboratory of Instituto de Investigación Biomédica de Málaga (IBIMA). Serum inflammatory biomarker levels (IL-6 and TNFa) and adipokines were measured using an enzyme-linked immunosorbent assay (ELISA) (R&D Systems, Inc., Minneapolis, MN, USA). For IL-6 levels, the minimum detectable concentration was 0.70 pg/mL. The intra- and inter-assay coefficients of variation were 2.6% and 4.5%, respectively. The minimum detectable concentration of TNFa levels, was 1.6 pg/mL. The intra- and inter-assay coefficients of variation were 4.7% and 5.8%, respectively. For adiponectin levels, the minimum detectable concentration was 0.246 ng/mL. The intra- and inter-assay coefficients of variation were 3.5% and 6.5%, respectively. For resistin levels, the minimum detectable concentration was 0.026 ng/mL. The intra- and inter-assay coefficients of variation were 4.7% and 8.4%, respectively. Lastly, high-ultrasensitivity CRP levels were measured using ELISA (DRG Instruments GmbH, Germany). The minimum detectable concentration was 0.1 mg/mL. The intra- and inter-assay coefficients of variation were 4.4% and 3.3%, respectively.

## Statistical analysis

Simple Interactive Statistical Analysis (SISA) was used to calculate the sample size. We assumed a 95% confidence level (0.5% error), a statistical power of 80%, and a loss rate of 5%. To detect differences in the inflammatory and adipokines profile, the sample size needed was 150 patients. Quantitative variables with normal distribution were expressed as mean ± standard deviation (SD) and qualitative variables were expressed as percentages. Student t-test to compare quantitative variables and the Chi-square test to compare qualitative variables were used. Bivariate correlations were determined using Pearson correlation coefficient. The SPSS program, version 22.0 for Windows (IBM Corporation INC. Somers, NY, USA) was used for the statistical analysis of results.

## Results

The initial recruitment from centers for the elderly included a total of 690 subjects who wanted to participate in the study. From these individuals, 524 did not meet the inclusion criteria or finally declined to participate. The final sample included 166 MHO elderly subjects (**Fig 1**).

From these 166 subjects, 40 (24.1%) were male and 126 (75.9%) female (p <0.0001), with an average age of 71.7±5.2 years (65 to 87 years old).

The anthropometric and clinical parameters, as well as MedDiet adherence at the initial visit and after 12 months of lifestyle modification, are shown in **Table 1**. After 12 months of intervention, the weight and BMI of our subjects remained similar with respect to baseline. However, their waist circumference decreased significantly in both male and female population (-2.6 cm for male and -3.7 cm for female after 12 months, p<0.0001 in both subgroups). Their SBP and DBP tended to decrease, although the difference was not statistically significant. Moreover, MedDiet adherence increased significantly (p<0.001) compared to the beginning of the intervention (9.4±2.0 vs 10.3±1.7 points). On the other hand, all intensity levels of PA decreased drastically and significantly respect to the baseline conditions (p<0.001). Light PA

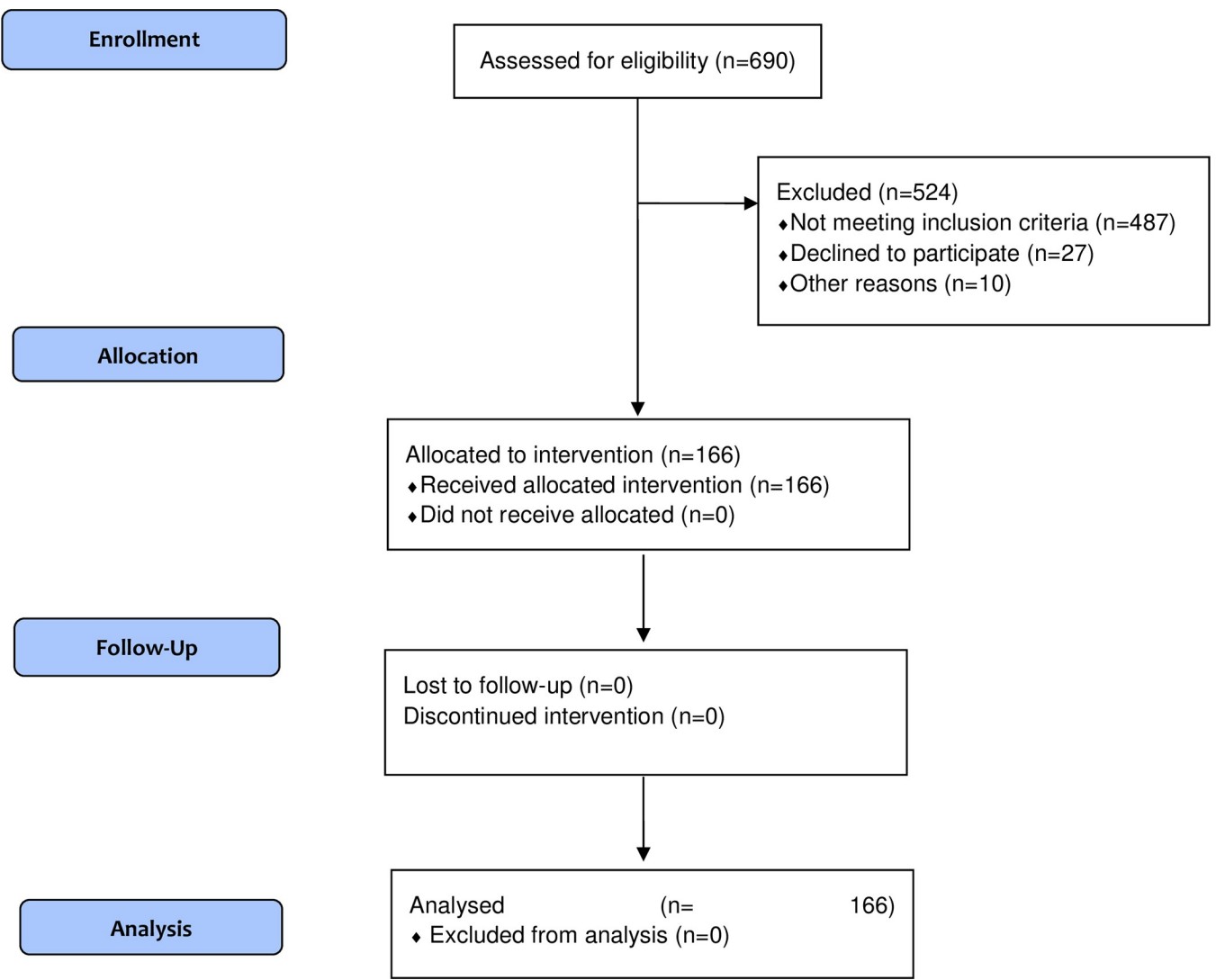

**Fig 1. Flow chart diagram of the selection patients in our analysis.**

**Table 1. Anthropometric parameters and MedDiet adherence at baseline and after 12 months of intervention (Mean±SD).**

|  | Baseline (B) | 12 months (12M) | p (B Vs. 12M) |
|---|---|---|---|
| Body weight (Kg) | 78.1±11.8 | 77.8±11.8 | 0.29 |
| BMI (kg/m$^2$) | 31.4±4.2 | 31.3±4.3 | 0.55 |
| Waist circumference (cm) | M 109.4±9.7 | 106.8±9.7 | <0.0001 |
|  | F 100.7±8.1 | 97.0±8.4 | <0.0001 |
| WHI | 0.9±0.1 | 0.9±0.1 | 0.01 |
| SBP (mmHg) | 136.6±17.7 | 136.2±18.4 | 0.81 |
| DBP (mmHg) | 85.6±10.3 | 84.5±8.7 | 0.31 |
| Adherence to MedDiet (points) | 9.4±2.0 | 10.3±1.7 | <0.001 |

BMI: Body Mass Index, M: Male. F: Female, WHI: Waist/Hip Index, SBP: Systolic Blood Pressure, DBP: Diastolic Blood Pressure, MedDiet: Mediterranean diet.

**Table 2. PA (minutes/day) at baseline and after 12 months of intervention (Mean±SD).**

|  | Baseline (B) | 12 months (12M) | p (B Vs. 12M) |
|---|---|---|---|
| **Light PA** | 710±142 | 262±84 | <0.001 |
| **Moderate PA** | 284±84 | 47±39 | <0.001 |
| **Vigorous PA** | 61±47 | 4±6 | <0.001 |
| **Moderate to vigorous PA** | 352±115 | 312±107 | <0.001 |

(-448 min/d), moderate (-237 min/d), vigorous (-57 min/d), and moderate-vigorous (-40 min/d), respectively. PA levels were summarized in **Table 2**.

Inflammatory biomarker concentrations are shown in **Table 3**. CRP serum levels and IL-6 decreased after 12 months of intervention, although their difference was not statistically significant. However, TNFa serum levels increased after the intervention (4.02±2.18 pg/mL at baseline vs. 6.81±3.43 pg/mL at 12 months, p<0.001). Concerning the adipokine profile, adiponectin serum levels increased after the intervention (1.05±5.19 ng/mL at baseline vs. 9.33±4.84 ng/mL at 12 months, p<0.001), whereas resistin serum levels decreased after 12 months of follow-up (8.29±3.45 ng/mL at baseline vs. 7.29±2.92 ng/mL at 12 months, p<0.001).

Correlation analysis (**Table 4**) in baseline conditions showed that only TNFa correlates with PA. Such biomarker presented a positive correlation with light PA (r = 0.19, p = 0.04) and negative correlation with moderate (r = -0.18, p = 0.05), moderate to vigorous (r = -0.2, p = 0.03), and vigorous PA (r = -0.22, p = 0.01). Adiponectin positively correlates with weight (r = 0.18, p = 0.04) and some PA levels, moderate (r = 0.31, p = 0.001) and moderate to vigorous PA (r = 0.26, p = 0.004), and negatively with light PA (r = -0.19, p = 0.04) and WHI (r = -0.29, p = 0.001). Finally, resistin had a positive correlation with weight (r = 0.19, p = 0.04) and light PA (r = 0.20, p = 0.03) and negative correlation with vigorous (r = -0.21, p = 0.02) and moderate to vigorous PA (r = -0.19, p = 0.04). After 12 months of intervention, CRP was positively associated with weight (r = 0.25, p = 0.01) and BMI (r = 0.31, p = 0.01), and negatively associated with WHI (r = -0.23, p = 0.02) were observed. Waist circumference (r = 0.22, p = 0.02), and moderate (r = -0.20, p = 0.04) and moderate-vigorous PA (r = -0.21, p = 0.03) correlate with IL-6 levels. In addition, TNFa only correlates negatively with weight (r = -0.19, p = 0.05). On the other hand, adiponectin concentration had a positive correlation (r = 0.24, p = 0.01) with WHI. Resistin levels positively correlate with weight (r = 0.22, p = 0.02) and waist circumference (r = 0.23, p = 0.02).

Regarding metabolic parameters, only adiponectin correlates negatively with insulin (r = -0.33, p<0.001) and triglycerides (r = -0.44, p<0.001) and positively with HDL-c (r = 0.53,

**Table 3. Adipokine and inflammatory biomarker levels at baseline and after 12 months of lifestyle modification (Mean±SD).**

|  | Baseline (B) | 12 months (12M) | p (B Vs. 12M) |
|---|---|---|---|
| **CRP (mg/L)** | 4.93±12.49 | 3.30±3.16 | 0.16 |
| **IL-6 (pg/mL)** | 1.71±1.84 | 1.60±1.55 | 0.52 |
| **TNFa (pg/mL)** | 4.02±2.18 | 6.81±3.43 | <0.001 |
| **Adiponectin (ng/mL)** | 1.05±5.19 | 9.33±4.84 | <0.001 |
| **Resistin (ng/mL)** | 8.29±3.45 | 7.29±2.92 | <0.001 |

CRP: C-reactive protein, normal values: 0.07–8.2 mg/L. IL-6: interleukin-6, normal values: 3.13–12.5 pg/mL. TNFa: tumor necrosis factor alpha, normal values < 15.6 pg/mL. Adiponectin, normal values 0.8–21 ng/mL. Resistin, normal values 6.1–26.4 ng/mL.

**Table 4. Pearson correlation coefficient: Inflammatory biomarkers and adipokines vs. anthropometric parameters, Mediterranean diet adherence, and physical activity intensity levels at baseline and after 12 months of intervention.**

| BASELINE | Weight | | BMI | | Waist | | WHI | | MedDiet Adherence | | Light | | Moderate | | Vigorous | | Moderate to Vigorous | |
|---|---|---|---|---|---|---|---|---|---|---|---|---|---|---|---|---|---|---|
| | | | | | | | | | | | **Physical Activity** | | | | | | | |
| | r | p | r | p | r | p | r | p | r | p | r | p | r | p | r | p | r | p |
| CRP | 0.01 | 0.89 | 0.05 | 0.55 | -0.03 | 0.72 | -0.06 | 0.48 | -0.11 | 0.22 | 0.02 | 0.82 | -0.02 | 0.87 | -0.05 | 0.56 | -0.04 | 0.66 |
| IL-6 | -0.01 | 0.96 | -0.03 | 0.77 | 0.01 | 0.88 | 0.05 | 0.58 | -0.13 | 0.15 | 0.05 | 0.58 | -0.11 | 0.24 | 0.09 | 0.34 | -0.04 | 0.67 |
| TNFa | -0.07 | 0.45 | -0.12 | 0.18 | -0.13 | 0.15 | -0.1 | 0.28 | 0.18 | 0.06 | **0.19** | **0.04** | **-0.18** | **0.05** | **-0.2** | **0.03** | **-0.22** | **0.01** |
| Adiponectin | **0.18** | **0.04** | 0.09 | 0.33 | -0.13 | 0.16 | **-0.29** | **0.001** | 0.07 | 0.44 | **-0.19** | **0.04** | **0.31** | **0.001** | 0.1 | 0.28 | **0.26** | **0,004** |
| Resistin | **0.19** | **0.04** | 0.08 | 0.41 | 0.13 | 0.14 | 0.01 | 0.93 | -0.05 | 0.61 | **0.20** | **0.03** | -0.12 | 0.18 | **-0.21** | **0.02** | **-0.19** | **0.04** |
| 12 months | Weight | | BMI | | Waist | | WHI | | MedDiet Adherence | | Light | | Moderate | | Vigorous | | Moderate to Vigorous | |
| | | | | | | | | | | | **Physical Activity** | | | | | | | |
| | r | p | r | p | r | p | r | p | r | p | r | p | r | p | r | p | r | p |
| CRP | **0.25** | **0.009** | **0.31** | **0.001** | 0.13 | 0.17 | **-0.23** | **0.02** | 0.01 | 0.95 | 0.09 | 0.38 | 0.05 | 0.64 | -0.08 | 0.4 | 0.08 | 0.44 |
| IL-6 | 0.14 | 0.16 | 0.08 | 0.40 | **0.22** | **0.02** | 0.11 | 0.26 | -0.02 | 0.84 | -0.18 | 0.07 | **-0.20** | **0.04** | -0.18 | **0.06** | **-0.21** | **0.03** |
| TNFa | **-0.19** | **0.05** | -0.04 | 0.7 | -0.08 | 0.43 | -0.09 | 0.38 | -0.07 | 0.51 | 0.18 | 0.06 | -0.03 | 0.75 | -0.04 | 0.65 | 0.12 | 0.22 |
| Adiponectin | -0.13 | 0.17 | 0.06 | 0.52 | -0.11 | 0.27 | **-0.24** | **0.01** | -0.09 | 0.4 | 0.14 | 0.15 | -0.07 | 0.47 | -0.03 | 0.72 | 0.08 | 0.44 |
| Resistin | **0.22** | **0.02** | 0.19 | 0.06 | **0.23** | **0.02** | 0.02 | 0.87 | -0.09 | 0.38 | -0.1 | 0.29 | -0.04 | 0.66 | -0.08 | 0.42 | -0.1 | 0.32 |

BMI: Body Mass Index, WHI: Waist/Hip Index, MedDiet: Mediterranean diet; PA: physical activity.

p<0.001) at baseline. After 12 months of intervention correlations remain similar; adiponectin correlates negatively with insulin (r = -0.20, p = 0.03) and triglycerides (r = -0.23, p = 0.01) and positively with HDL-c (r = 0.30, p = 0.001).

In baseline conditions, resistin levels correlate positively with TNFa (r = 0.25, p = 0.01) and CRP (r = 0.31, p<0.0001) levels. TNFa and IL-6 correlates positively with CRP (r = 0.19, p = 0.03 and r = 0.36, p<0.0001, respectively). After 12 months of intervention, only IL-6 correlates positively with CRP (r = 0.26, p = 0.006). Data indicate that adipokines levels correlate positively during the process of lifestyle modification (adiponectine: basal vs 12 months, r = 0.75, p<0.0001 and resistin: basal vs 12 months, r = 0.74, p<0.0001). However, during this process of lifestyle modification, only IL-6 correlated positively with itself (basal vs 12 months, r = 0.46, p<0.0001) and with CRP (Il-6 basal vs CRP 12 months, r = 0.25, p = 0.03).

## Discussion

This study was affected by the COVID-19 pandemic situation. In this context, our MHO elderly population had to experience a strict and extensive lockdown. Their practice of PA obviously decreased due to the epidemiologic circumstances. Older adults are more vulnerable to COVID-19 [18], as health authorities, politicians, and social media have widely reported. Besides the restrictions, the infection fear has decreased PA in older people [19] and it has also affected our study results. Moreover, dietary habits have changed during lockdown in many people, with a higher frequency of food, snacking, and alcoholic drinks consumption [20]. This trend towards unhealthy food habits during lockdown, along with a decreased PA due to mobility restrictions, has led to weight gain in many cases, especially in already people with obesity [21]. Despite this scenario, our MHO elderly participants maintained their healthy lifestyle, even increasing their adherence to MedDiet so that their weight and BMI remained similar to the baseline values.

It is noted that the female gender is the main group in our study. Women are reported to participate more in this type of health improvement studies, which leads to a longer life expectancy than men. In elderly ages, the survival of women (with a mean life of 87 years old) is more frequent compared to men (with a life expectancy of 83.6 years old) in our environment (Málaga, Spain) [22]. Women are more concerned about the importance of healthy habits in order to prevent metabolic diseases.

Previous studies have demonstrated that participants who underwent an intensive lifestyle modification, with weight control and practice of PA, improved their levels of adipokines and cytokines, not only during the intervention period, but these changes persisted for 12 months post-intervention [23]. Inflammatory profile in our MHO elderly subjects was in normal range both in baseline conditions and after 12 months of lifestyle modification. CRP and IL-6 decreased after intervention, data in concordance with Gokulakrishnan K et al. [23]. CRP is a pentameric protein with hepatic origin related to inflammation process and obesity. It is demonstrated that CRP is associated with BMI [24]. When the inflammation cascade is activated, T cells and macrophages secrete another inflammatory biomarker, IL-6. It is known that IL-6 levels increase with adiposity, and that 15–25% of circulating IL-6 might be released by adipose tissue in vivo [25]. Although both IL-6 and TNFa are expressed by adipose tissue, there are important differences in their systemic release. IL-6 is released from subcutaneous adipose tissue. However, TNFa is not released by this type of adipose tissue. Although our MHO elderly population had a very low weight loss, our results showed significantly decreased levels of inflammatory markers (CRP and IL-6) after intervention. Other studies have suggested that lifestyle modifications following a healthy diet and regular practice of PA decrease plasma levels of inflammatory cytokines as IL-6 and TNFa [26]. However, TNFa concentrations experience a significant increment, although their values are in normal range after 12 months of intervention, perhaps due to the advanced age of our population. In elderly population, sarcopenic/cachectic obesity can occur, with a permanent loss of skeletal muscle mass (sarcopenia), with or without loss of fat mass [27]. This is the case of some patients with chronic inflammatory diseases of the joints, in whom the effect of the increased metabolism induced by activation of immune response [28] is compensated with minor practice of PA. He J. et al. demonstrated that TNFa expression in adipose tissue is positively associated with 24-h energy expenditure [29]. These data are in agreement with our results, in which TNFa is increased when our elderly MHO participants reduced their practice of PA in all their levels (moderate or vigorous PA). Regarding correlation between inflammation biomarkers and PA, only TNFa correlated with PA in baseline conditions. It showed negative correlation with moderate, moderate to vigorous and vigorous PA. As it has been demonstrated by other authors, TNFa tends to decrease with PA, but especially with PA of moderate intensity [30]. Our subjects had a high level of moderate and moderate to vigorous PA at baseline, but after 12 months of intervention, there was no correlation between PA and TNFa as PA drastically decreased due to COVID-19 restrictions.

Our population presented a normal range of adipokine profile (adiponectin and resistin) in pre and post intervention conditions. High concentrations of adiponectin are associated with a higher adherence to the MedDiet in adult population [31]. This fact has been corroborated in our study, despite the fact that there is a lack of a statistically-significant correlation. MedDiet with weight loss significantly increased plasma adiponectin levels [32]. Adipose tissue is an important endocrine organ. The adipocytes secrete multiple, metabolically active proteins, adipokines. Adiponectin is secreted by adipocytes, and paradoxically, its levels are decreased in subjects with obesity [33]. Adiponectin is mostly expressed in subcutaneous adipose tissue. Its expression and blood concentration decrease as adiposity increases and low levels of adiponectin are associated with disease states such as diabetes and cardiovascular disease. Our

elderly population increased significantly their levels of adiponectin despite their BMI being maintained. However, another adipokine, resistin, decreases significantly and correlates positively with weight and abdominal circumference in our elderly MHO population. Resistin is an adipokine linked to obesity and diabetes and involved in the development of insulin resistance. Normally, the serum concentration of resistin in humans ranges from 7 to 22 ng/mL. Resistin activates the transcription of pro-inflammatory genes and cytokines as IL-6 and TNFa, through activation of transcription factor, nuclear factor kappa B (NF-κB) [34]. In studies of subjects with obesity, high resistin concentrations correlate with weight and BMI [35]. Diet and PA decrease resistin levels, which is typically accompanied by a reduction in BMI [36], in concordance with data found in this study. On the other side, high-intensity PA induced an improvement in inflammatory biomarkers and insulin resistance, with a reduction in IL-6, TNFa, CRP, and resistin and an increase in adiponectin, thus indicating that exercise exerts anti-inflammatory and insulin-sensitising effects [37]. These data are in concordance with the data reported in this study.

## Limitations

An important limitation in this study could be the lack of a control group, but it is important that each participant was aware of his/her progression and improvement of his/her life quality when they follow a healthy lifestyle. For this reason, a comparison of inflammatory and adipokine profile in baseline conditions and after 12-month lifestyle intervention could note the efficacy of intervention.

Other limitations of our study are, firstly, the difference between the number of male and female participants. It has been demonstrated than women are more participative in lifestyle modification studies than male subjects, as they are usually more concerned about health care. And secondly, our study was performed on a Caucasian population sample, and therefore it cannot be extrapolated to other ethnic groups.

## Conclusions

In conclusion, aging is a complex and multifactorial biological process with an increased prevalence of metabolic abnormalities such as obesity. Adipocytes modulate the activity of pre-adipocytes, endothelial cells, and monocytes/macrophages due to increased production of inflammatory cytokines (CRP, IL-6, TNFa) and adipokines (adiponectin, resistin) among others. In addition, age-related body composition changes could play a role in the expression of inflammatory biomarkers and adipokines. As this study demonstrated a personalized MedDiet and PA program could modify and improve this process.

## Supporting information

**S1 Protocol.**
(DOCX)

## Acknowledgments

We would like to thank to the Sports Area (Sport Medicine) of Malaga City Hall (Andalusia, Spain) for allowing us and facilitating access to the different centers for the healthy elderly, and to Ms. Claudia Corazza González for her help with the final English-language version.

## Author Contributions

**Conceptualization:** Lidia Cobos-Palacios, María Isabel Ruiz-Moreno, Mónica Muñoz-Úbeda, Ricardo Gomez-Huelgas, María Rosa Bernal-Lopez.

**Data curation:** María Isabel Ruiz-Moreno, Antonio Vargas-Candela, Mónica Muñoz-Úbeda, Juan José Mancebo-Sevilla.

**Formal analysis:** María Isabel Ruiz-Moreno, Mónica Muñoz-Úbeda, Javier Benítez Porres, Ana Navarro-Sanz, Juan José Mancebo-Sevilla, María Rosa Bernal-Lopez.

**Funding acquisition:** Ricardo Gomez-Huelgas, María Rosa Bernal-Lopez.

**Investigation:** María Isabel Ruiz-Moreno, Mónica Muñoz-Úbeda, María Rosa Bernal-Lopez.

**Methodology:** Lidia Cobos-Palacios, María Isabel Ruiz-Moreno, Alberto Vilches-Perez, Javier Benítez Porres, Ana Navarro-Sanz, María Dolores Lopez-Carmona, Jaime Sanz-Canovas.

**Project administration:** María Rosa Bernal-Lopez.

**Resources:** María Isabel Ruiz-Moreno, Mónica Muñoz-Úbeda, Javier Benítez Porres, Ana Navarro-Sanz.

**Software:** Alberto Vilches-Perez, Antonio Vargas-Candela, Javier Benítez Porres, María Dolores Lopez-Carmona, Jaime Sanz-Canovas, Luis M. Perez-Belmonte.

**Supervision:** Ricardo Gomez-Huelgas, María Rosa Bernal-Lopez.

**Validation:** Lidia Cobos-Palacios, María Dolores Lopez-Carmona, Jaime Sanz-Canovas, Luis M. Perez-Belmonte.

**Visualization:** Ricardo Gomez-Huelgas, María Rosa Bernal-Lopez.

**Writing – original draft:** Lidia Cobos-Palacios, María Isabel Ruiz-Moreno, Mónica Muñoz-Úbeda, María Rosa Bernal-Lopez.

**Writing – review & editing:** Lidia Cobos-Palacios, María Isabel Ruiz-Moreno, María Rosa Bernal-Lopez.

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
