## [Decision Letter · Decision Letter 0]

5 Jan 2022

PONE-D-21-33113Metabolically Healthy Obesity: Inflammatory biomarkers and adipokines in elderly populationPLOS ONE

Dear Dr. Bernal-Lopez, Thank you for submitting your manuscript to PLOS ONE. After careful consideration, while the reviewers thought the research is of interest, we feel that it has merit but does not fully meet PLOS ONE’s publication criteria as it currently stands. Therefore, we invite you to submit a revised version of the manuscript that addresses the points raised during the review process.  Please read carefully the comments from both reviewers and address their questions one-by-one.  Please submit your revised manuscript by Feb 11 2022 11:59PM. If you will need more time than this to complete your revisions, please reply to this message or contact the journal office at plosone@plos.org. Please include the following items when submitting your revised manuscript:A rebuttal letter that responds to each point raised by the academic editor and reviewer(s). You should upload this letter as a separate file labeled 'Response to Reviewers'.A marked-up copy of your manuscript that highlights changes made to the original version. You should upload this as a separate file labeled 'Revised Manuscript with Track Changes'.An unmarked version of your revised paper without tracked changes. You should upload this as a separate file labeled 'Manuscript'.

We look forward to receiving your revised manuscript.

Kind regards,

Kai Sun, MD, PhD

Associate Professor

Academic Editor

PLOS ONE

Journal Requirements:

No

This work was supported by grants from the Instituto de Salud Carlos III, cofunded by the Fondo Europeo de Desarrollo Regional-FEDER (“Centros de Investigación En Red” (CIBER, CB06/03/0018) and PI18/00766. M Rosa Bernal-Lopez was supported by “Miguel Servet Type II” program (CPII/00014) and “Nicolas Monardes” program (C1-0005-2020), Lidia Cobos Palacios and Jaime Sanz Cánovas were supported by “Rio Hortega” program (CM20/00125 and CM20/00212, respectively) from the ISCIII-Madrid (Spain), cofunded by the Fondo Europeo de Desarrollo Regional-FEDER. Monica Muñoz Ubeda was supported by Consejeria de Salud, Junta de Andalucía (RH-0100-2020)

No

5. Please ensure that you refer to Figure 1 in your text as, if accepted, production will need this reference to link the reader to the figure.

6. We noticed you have some minor occurrence of overlapping text with the following previous publication(s), which needs to be addressed:

- https://doi.org/10.1016/j.mad.2019.01.004

The text that needs to be addressed involves the Conclusion.

In your revision ensure you cite all your sources (including your own works), and quote or rephrase any duplicated text outside the methods section. Further consideration is dependent on these concerns being addressed.

Reviewers' comments:

Reviewer's Responses to Questions

**Comments to the Author**

1. Is the manuscript technically sound, and do the data support the conclusions?

Reviewer #1: Partly

Reviewer #2: Partly

2. Has the statistical analysis been performed appropriately and rigorously? 

Reviewer #1: Yes

Reviewer #2: No

3. Have the authors made all data underlying the findings in their manuscript fully available?

Reviewer #1: Yes

Reviewer #2: Yes

4. Is the manuscript presented in an intelligible fashion and written in standard English?

Reviewer #1: Yes

Reviewer #2: Yes

5. Review Comments to the Author

Reviewer #1: In this paper, the authors aimed to examine the impact of lifestyle modifications by following a Mediterranean Diet (MedDiet) program and physical activity (PA) training on inflammatory biomarkers and adipokine profile in a Metabolically Healthy Obese (MHO) elderly population. 166 MHO elderly subjects including 40 males and 126 females were included. They observed that 12 months of intervention reduced the waist circumference but did not change body weight and BMI. MedDiet adherence increased, but all intensity levels of PA decreased. While CRP and IL-6 tended to be reduced, TNFa was increased after 12 months, when adiponectin level was also elevated and resistin concentrations decreased. TNFa, adiponectin, and resistin correlated with PA at baseline and after 12 months and CRP, IL-6, TNFa, adiponectin, and resistin concentrations correlated with anthropometric parameters and some intensities of PA. At baseline, resistin levels correlated positively with TNFa and CRP levels. TNFa and IL-6 correlated positively with CRP. After 12 months, only IL-6 correlated positively with CRP.

This paper reported some interesting correlations as analyzed and presented from a small cohort of elderly MHO subjects, and was generally well written, but provided limited novel information. The main issue is that without a control group, how the authors can tell the changes at 12 months were because of interventions rather than a natural aging process. The significance of the correlations is unknown. While correlations of the markers with some anthropometric parameters and PA intensities were analyzed, their relations to metabolic parameters such as glucose, insulin, and lipid levels were not analyzed. Also, the conclusion of the study was not clearly stated.

Reviewer #2: The authors of this study explored the phenotype of metabolically healthy obesity. Multiple biomarkers of obesity and inflammation were assessed in response to lifestyle modifications. In my opinion, the authors do not use appropriate statistical methods in the study (e.g, Student t-test, assessments of normal distributions). A revision must include justification of statistical tests and apply post-hoc testing.

6. PLOS authors have the option to publish the peer review history of their article (what does this mean?). If published, this will include your full peer review and any attached files.

Reviewer #1: No

Reviewer #2: No

---

## [Author Response · Author response to Decision Letter 0]

11 Feb 2022

PONE-D-21-33113

Metabolically Healthy Obesity: Inflammatory biomarkers and adipokines in elderly population

PLOS ONE

Dear Dr. Sun: 

We thank you very much for giving us the opportunity to revise our manuscript. We have carefully considered all comments. Each comment has been addressed and we have modified the manuscript accordingly. 

We sincerely hope that the current version of the manuscript will be acceptable for publication in your journal. 

All changes are shown in red so that they may be easily seen. 

The response to the reviewers are included as separate files in the web page

Dr. Maria Rosa Bernal-Lopez

---

## [Decision Letter · Decision Letter 1]

1 Mar 2022

Metabolically Healthy Obesity: Inflammatory biomarkers and adipokines in elderly population

PONE-D-21-33113R1

Dear Dr. Bernal-Lopez,  

We’re pleased to inform you that your manuscript has been judged scientifically suitable for publication and will be formally accepted for publication once it meets all outstanding technical requirements.

Kind regards,

Kai Sun, MD PhD

Associate Professor

Academic Editor

PLOS ONE

Additional Editor Comments (optional):

Reviewers' comments:

Reviewer's Responses to Questions

**Comments to the Author**

1. If the authors have adequately addressed your comments raised in a previous round of review and you feel that this manuscript is now acceptable for publication, you may indicate that here to bypass the “Comments to the Author” section, enter your conflict of interest statement in the “Confidential to Editor” section, and submit your "Accept" recommendation.

Reviewer #1: All comments have been addressed

Reviewer #2: All comments have been addressed

2. Is the manuscript technically sound, and do the data support the conclusions?

Reviewer #1: Yes

Reviewer #2: Yes

3. Has the statistical analysis been performed appropriately and rigorously? 

Reviewer #1: Yes

Reviewer #2: Yes

4. Have the authors made all data underlying the findings in their manuscript fully available?

Reviewer #1: Yes

Reviewer #2: Yes

5. Is the manuscript presented in an intelligible fashion and written in standard English?

Reviewer #1: Yes

Reviewer #2: Yes

6. Review Comments to the Author

Reviewer #1: Thanks the authors for the responses. I do not have further comments on this manuscript. it is acceptable to this reviewer.

Reviewer #2: (No Response)

7. PLOS authors have the option to publish the peer review history of their article (what does this mean?). If published, this will include your full peer review and any attached files.

Reviewer #1: No

Reviewer #2: No

---

## [Editor Report · Acceptance letter]

19 May 2022

PONE-D-21-33113R1 

Metabolically Healthy Obesity: Inflammatory biomarkers and adipokines in elderly population 

Dear Dr. Bernal-Lopez:

I'm pleased to inform you that your manuscript has been deemed suitable for publication in PLOS ONE. Congratulations! Your manuscript is now with our production department. 

Kind regards, 

on behalf of

Dr. Kai Sun 

Academic Editor

PLOS ONE